# Single Teeth and Partial Implant Rehabilitations Using Ultra-Hydrophilic Multi-Zone Anodized Surface Implants: A Retrospective Study with 1-Year Follow-Up

**DOI:** 10.3390/jcm14010066

**Published:** 2024-12-26

**Authors:** Miguel de Araújo Nobre, Carolina Antunes, Ana Ferro, Armando Lopes, Miguel Gouveia, Mariana Nunes, Diogo Santos

**Affiliations:** 1Research, Development and Education Department, MALO CLINIC, Avenida dos Combatentes, 43, Level 11, 1600-042 Lisboa, Portugal; cantunes@maloclinics.com; 2Oral Surgery Department, MALO CLINIC, Avenida dos Combatentes, 43, Level 9, 1600-042 Lisboa, Portugal; aferro@maloclinics.com (A.F.); alopes@maloclinics.com (A.L.); mgouveia@maloclinics.com (M.G.); mnunes@maloclinics.com (M.N.); dsantos@maloclinics.com (D.S.)

**Keywords:** dental implants, bone resorption, osseointegration, single-tooth implants, cumulative survival rate, implant-supported dental prosthesis

## Abstract

**Background/Objectives**: In the last decades, dental implant surfaces have been evolving to increase success and implant survival rates. More studies evaluating outcomes with implants with ultra-hydrophilic multi-zone anodized surfaces are necessary. The aim of this study is to evaluate the short-term outcome of implants of conical connection with anodized ultra-hydrophilic surfaces for support of single teeth and partial rehabilitations. **Methods**: In this retrospective study, patients received parallel-walled implants with a gradually anodized surface. The primary outcome measure was implant survival. Secondary outcome measures were marginal bone loss and mechanical and biological complications. This study included 253 conical connection implants with anodized ultra-hydrophilic surfaces, placed in 145 patients (71 males and 74 females; average age: 55.8 years). Sixty patients presented comorbidities, and 19 patients presented smoking habits. **Results**: Ten patients (15 implants) were lost to follow-up. Two implants failed in two patients, resulting in a cumulative survival rate of 99.2%, with 98.5% and 100% for males and females, respectively, and 99.1% and 100% for single teeth and partial rehabilitations, respectively. The average marginal bone loss was 0.52 mm at 1 year, with 0.60 mm and 0.42 mm for males and females, respectively, and 0.52 mm and 0.50 mm for single teeth and partial rehabilitations, respectively. The rate of mechanical complications was 4.8% and 3.2% at patient and implant levels, respectively. Biological complications occurred in one patient (0.7%) at one implant (0.4%). **Conclusions**: These results indicate that the use of implants with ultra-hydrophilic multi-zone anodized surfaces for single teeth and partial rehabilitations is viable in the short term.

## 1. Introduction

Over the last two decades, dental implant surfaces have been evolving to increase success and implant survival rates [1,2]. These changes included implant surface treatments, aiming to improve the osseointegration process, decrease the chances of infection, and promote cell growth and proliferation [2].

Machined titanium surface implants with an untreated surface were initially used [1]. However, the results achieved made clear that there was a need for improved osseointegration and, consequently, a stronger primary stability, leading to the possibility of early implant loading [1]. To achieve this, surface roughness modifications were carried out, with the anodization process (TiUnite, Nobel Biocare AB, Gothenburg, Sweden) as one of the examples [1]. Despite the improvement in implant success for implants with anodically oxidized surfaces compared to machined surfaces [1], biological complications associated with anodically oxidized surface implants remained significant [3]. Although it is proven that an implant surface with an increased roughness represents a benefit during the osseointegration period, some studies suggest that this roughness may play a decisive role in the development of peri-implantitis once it becomes exposed to the oral cavity [4,5,6]. When compared with machined surfaces, anodically oxidized surfaces demonstrated overall higher survival rates but also a higher incidence of peri-implant pathology, with biofilm adhesion and maturation tending to be slower on machined surfaces [4,7]. Additionally, a rougher surface may also increase the release of titanium particles during implant insertion [8], potentially leading to an inflammatory process in the peri-implant tissues since titanium particles increase bacterial biofilm virulence [8,9,10]. Aiming for an increase in the quality of soft tissue adhesion and corresponding soft tissue outcomes, a new surface concept was developed consisting of an ultra-hydrophilic surface with gradual anodization, with increasing roughness and oxide layer thickness from the implant platform to the apex (TiUltra, Nobel Biocare AB) and a minimally rough surface for the abutments (Xeal, Nobel Biocare AB) [11].

Recent studies registered high survival rates for implants and abutments with ultra-hydrophilic surfaces, potentially providing the clinical and biological features necessary at every tissue level. Ongoing studies should evaluate the success of these new anodized surface implants [11,12,13].

In a previous study, implants with ultra-hydrophilic multi-zone anodized surfaces were used for support of full-arch restorations, registering a 100% prosthetic and implant cumulative survival rates with an average follow-up of 23 months, together with a 1-year average marginal bone loss of 0.39 mm, and absence of biological complications [14]. Another investigation, consisting of a multicenter evaluation of 916 patients, rendered a 99.1% cumulative implant survival rate at 5 months [15]. Therefore, more human studies are necessary to investigate the outcome of dental implants with ultra-hydrophilic multi-zone anodized surfaces. The aim of this study is to evaluate the short-term outcome of implants of conical connection with anodized ultra-hydrophilic surfaces with a gradual topography from the collar to the apex for support of single teeth and partial rehabilitations.

## 2. Materials and Methods

### 2.1. Setting

This article was written following the STROBE (Strengthening the Reporting of Observational Studies in Epidemiology) guidelines [16]. This retrospective study comprises data regarding the use of parallel-walled implants with anodized ultra-hydrophilic surfaces with a gradual topography from the collar to the apex (NobelParallel Conical Connection TiUltra, Nobel Biocare AB) for single teeth and partial bridge rehabilitations. This study was carried out at a private clinic in Lisbon, Portugal, and was approved by an independent ethical committee (Ethics Committee for Health, authorization no. 001/2023, date of approval: 25 September 2023). All patients provided written informed consent in order to participate in this investigation.

### 2.2. Inclusion and Exclusion Criteria

Inclusion criteria were patients in need of single or partial rehabilitations in both arches, with enough bone height and width availability to place at least 7 mm long implants, sufficient implant primary stability, and those who were followed exclusively at the clinic. Patients of both genders and any age were considered. Exclusion criteria included patients who refused to give consent to participate in the study, patients rehabilitated with implants inserted in grafted bone, patients in need of full-arch rehabilitations and patients with single teeth and partial rehabilitations with other implant systems. Patients were followed for one year.

### 2.3. Surgical Protocol

The surgical procedure was performed under local anesthesia with mepivacaine chlorhydrate and adrenalin 1:100,000 (Scandinibsa 2%^®^, Inibsa Laboratory, Barcelona, Spain). In the event of anxiety, patients were sedated with diazepam (Valium^®^ 10 mg, Roche, Amadora, Portugal) before surgery. Antibiotics (amoxicillin 875 mg + clavulanic acid 125 mg, Labesfal, Campo de Besteiros, Portugal) were given 1 h before surgery and on a daily basis for 6 days. Anti-inflammatory medication (ibuprofen, 600 mg, Ratiopharm, Lda, Carnaxide, Portugal) was administered for 6 days postoperatively. Analgesics (clonixine [Clonix^®^, Janssen-Cilag Farmaceutica, Lda, Barcarena, Portugal], 300 mg) were given on the day of surgery and postoperatively if necessary. Antacid medication (omeprazole, 20 mg, Lisboa, Portugal) was given on the day of surgery and daily for 6 days postoperatively. For the flap procedure used in the majority of patients, the placement of implants followed the standard procedures, with the following adjustments: for maximum tissue repositioning of the vestibular aspect of the flap, the incision was performed on the palatal side of the crest, with two parallel discharges towards the vestibulum. For both techniques, the drilling sequence was adjusted to achieve maximum bone compression and anchorage, with the implant sites initially prepared with 2.0-mm twist drills, followed by cortical bone enlargement with 3.2- or 3.6-mm step drills and elimination of countersinking for marginal bone preservation. The implants placed were 3.75 mm or 4.3 mm in width. It was intended to place the implant platform flush to the bone crest. Bicortical anchorage was performed whenever possible.

### 2.4. Prosthodontic and Maintenance Protocols

The abutment choice was made according to the rehabilitation. For single teeth, multiunit or immediate healing abutments (Nobel Biocare AB) were used, depending on whether the immediate prosthesis was screw-retained or cement-retained, respectively. For small fixed dental prostheses, multiunit abutments were used (Nobel Biocare AB). A provisional acrylic resin crown or fixed dental prosthesis was manufactured and attached to the implants on the same day as the surgery, achieving immediate implant function.

Patients were enrolled in an implant maintenance program. Ten days post-surgery, sutures were removed (for patients who underwent the flap surgery), and implant stability was evaluated manually by applying lateral movements to the implant. This procedure was then repeated at 2 and 4 months post-surgery until a stable condition was achieved. Final prostheses were delivered at 6 months, consisting of ceramic crowns or ceramic/metal–ceramic fixed dental prostheses. Figure 1, Figure 2, Figure 3, Figure 4, Figure 5, Figure 6 and Figure 7 illustrate a clinical case from the present study, with the insertion of two parallel-walled implants in the 2nd quadrant.

### 2.5. Outcome Measures

The primary outcome measure was implant survival. Secondary outcome measures were marginal bone loss and mechanical and biological complications. Implant survival was based on function. Marginal bone loss was assessed by examining periapical radiographs of the implants. The reference point for the reading was the implant platform (the horizontal interface between the implant and the abutment), and marginal bone loss was defined as the difference in marginal bone level between 1-year follow-up and the time of surgery. Radiographs were taken using a parallel technique with a film holder (Super-bite, Hawe-Neos, Switzerland) and an aiming device. Radiographs were accepted or rejected based on the clarity of the implant threads. The assessment of mechanical complications included fracture or loosening of prosthetic or abutment screws. The assessment of biological complications included infection, fistula, or abscess formation.

### 2.6. Statistical Analysis

Lifetables were used to estimate implant survival. Descriptive statistics were performed in order to evaluate the secondary outcomes of this study, such as the marginal bone loss and the rates of mechanical and biological complications. Inferential statistics were performed to compare the distribution between genders for the variables “implant survival” and “marginal bone loss”. The variable “implant survival was dichotomized (0: survival; 1: failure) and analyzed using the chi-square test and, assuming the applicability conditions, the Fisher Exact test. The variable “marginal bone loss” was first evaluated considering its distribution using the Kolmogorov–Smirnov test, and then the difference in distribution between genders evaluated using the non-parametric test Mann-Whitney U. Significance level was set at 5%. Statistics were computed using IBM SPSS Statistics 26.

## 3. Results

The study included 145 consecutive patients (71 males and 74 females) with an average (standard deviation) age of 55.8 years (12.7), ranging from 20 to 88. The patients were followed for 1 year (mean: 11.7 months). Nineteen patients (13.1%) were smokers, and 60 patients (41.4%) presented systemic conditions, of which 5 (3.5%) presented bruxism. Thirty-eight patients (26.2%) presented a single systemic condition, while 22 (15.2%) presented two or more. Of the patients with systemic diseases, 4 had infectious diseases, 1 had an oncologic problem, 1 had a blood disease, 1 had an immunologic problem, 3 had endocrine diseases, 2 had mental health problems, 2 had sleep disorders, 5 had nervous system diseases, 1 had a visual problem, 25 had circulatory system diseases, 2 had respiratory system diseases, 8 had digestive system diseases, 1 had a renal problem, 8 had musculoskeletal diseases, 3 had anxiety, 1 had postsurgical pain, 18 had factors that influenced health status, including smoking, and 1 had sinusitis.

Two hundred and fifty-three parallel-walled implants with gradually anodized surfaces were placed (119 in the maxilla and 134 in the mandible) between April 2019 and January 2021 (Table 1).

From these, 246 implants (97.2%) were placed using an open flap and seven implants (2.8%) through a flapless surgical procedure using a soft tissue punch in three implants (n = 3 patients) and into post-extraction sockets in four implants (n = 3 patients). Fifty-six (22.1%) implants were placed in immediate function, while the rest were placed in a two-staged protocol. The implants supported 218 single teeth and 21 partial bridge rehabilitations. The dimensions of the implants used in this study are presented in Table 2.

In 161 implants, a healing abutment was placed after surgery and in 92 implants, a MultiUnit abutment was used, of which 24 had anodized surfaces and 68 had machined surfaces.

Ten patients (6.9%) with 15 implants (5.9%) were lost to follow-up at 1 year. All these patients became unreachable (Table 3). At 1-year follow-up, two implants (0.8%) in two patients (1.4%) were lost, giving a cumulative survival rate of 99.2% at the implant level (Table 3). No significant differences were registered between genders (*p* = 0.0235, Fisher exact test).

The two failed implants were lost on the first 2 months of follow-up, as they did not osseointegrate. One of the implants was replaced (implant not accounted for in this study). Both implants that failed had a 4.3 mm diameter and 10 mm length and were both placed on the posterior mandible with a flap procedure, supporting single tooth rehabilitations. These implants were placed in two patients, one healthy patient and one patient presenting a comorbidity (immune thrombocytopenia); both patients did not present smoking nor bruxing habits, with no registration of mechanical nor biological complications in these implants prior to failure.

Concerning the marginal bone loss, from the 236 implants that reached the 1-year follow-up, 201 (85.2%) had readable radiographs. The average (standard deviation) marginal bone loss at 1 year was 0.52 mm (0.52 mm) (Table 4). No significant differences were registered between genders (*p* = 0.140, Mann–Whitney U test).

Eight implants (3.2%) in seven patients (4.8%) registered mechanical complications. These complications included crown fracture (n = 4), prosthetic screw loosening (n = 2 single tooth restoration), prosthesis fracture (n = 1 partial restoration), and abutment screw loosening (n = 1 single tooth restoration). All mechanical complications occurred in provisional prostheses. All situations were resolved: three complications were resolved by retightening the abutment and prosthetic screws during the appointment; three of the crown fractures were fixed during the appointment; one crown fracture was resolved by replacing the crown; and one prosthesis fracture was fixed in the laboratory, with all patients having the occlusion adjusted. Biological complications occurred at one implant (0.4%) and one patient (0.7%) during the 1-year follow-up. The implant presented an infection, with a probing pocket depth of 6 mm, that was resolved through non-surgical intervention (scaling, prophylaxis, and pocket irrigation with 0.2% chlorhexidine gel).

## 4. Discussion

The results of the present study suggest that the use of parallel-walled implants with gradually anodized surfaces is successful in the short term with low rates of complication and low levels of marginal bone loss. The main outcome of the present study indicates an implant cumulative survival rate of 99.2% at 1 year for single-tooth and partial implant rehabilitations. Two implants were lost in two patients, one of them presenting immune thrombocytopenia, a hemorrhagic condition characterized by low platelet count [16]. Since platelets act in blood clotting and bleeding prevention, this condition may have potentially interfered with the biological osseointegration process [17].

Although anodized surface implants represented a better option than machined surfaced ones, the need for implants that provided specific clinical and biological features in the different tissues led to the development of gradually anodized implants with an ultra-hydrophilic anodized surface [13]. This type of implant surface presents an increasing roughness and oxide layer thickness from the implant platform to the apex, providing the necessary roughness in the apex zone for successful osseointegration of the implant and the required characteristics in the implants’ coronal third (the implant platform) to facilitate the soft tissue adhesion and establish a stable peri-implant complex to decrease the probability of biological complications occurrence [13]. A previous long-term study of anodically oxidized surface implants in immediate function concluded that the cumulative survival rate was 98.1% and 95.2% at 1 and 12 years, respectively [18]. Considering short-term evaluations, two studies reported the outcome of implants with anodically oxidized surface, one of short implants in posterior jaws reporting a cumulative survival rate of 95.3% at 1-year follow-up [19], and another concerning single tooth rehabilitations registering a 98.1% cumulative survival rate at 1 year [20]. The results from both studies reveal a lower cumulative survival rate compared to the present study at 1-year follow-up. The lower survival rate in the study evaluating short implants in posterior jaws [19] could potentially be explained by the exclusive use of short implants in areas with high loading demands.

Concerning implants with ultra-hydrophilic anodized surfaces, Pozzi et al. [21] studied the success of parallel-walled implants in immediate loading after one year. They reported no implant failures nor mechanical or biological complications [21]. The marginal bone loss reported was −0.72 (0.26 mm) [21]. These results are comparable to the ones in the present study; however, in this study [21], patients with severe systemic conditions, under bisphosphonates therapy, heavy smokers and moderate to severe bruxers were excluded, conditions that may affect implant success. Additionally, two other previous studies investigated the success of ultra-hydrophilic multi-zone anodized surface implants in immediate function but in full arch rehabilitations [14,22]. Pozzi et al. [22] reported, in a prospective study with a mean follow-up period of 16 months, a cumulative survival rate of 98.3% and a mean (standard deviation) marginal bone loss of 0.53 mm (0.28 mm) for ultra-hydrophilic multi-zone anodized surface implants. Ferro and De Araújo Nobre reported 100% implant survival after a mean follow-up of 23 months and a mean (standard deviation) marginal bone loss of 0.39 mm (0.51 mm) [14]. The results of these studies [14,22] reveal similar survival rates to the present study nevertheless the smaller sample sizes [14,22]. Considering it is a recent technology and despite the low number of studies, the results point to high implant survival rates when using implants with an ultra-hydrophilic multi-zone anodized surface, suggesting a high treatment predictability.

In the present study, a mean marginal bone loss (standard deviation) of 0.52 mm (0.52 mm) at 1-year follow-up was reported. The differences in the marginal bone loss distribution between men and women and between partial rehabilitations and single teeth rehabilitations were clinically negligible [23]. The results compare favorably to other studies using anodically oxidized surface implants [19,24,25,26] or machined surface implants [27] in the same indication (immediate loading of dental implants for single and partial rehabilitations). A previous study reported a mean (standard deviation) marginal bone loss of 1.27 mm (0.67 mm) at 1-year follow-up; however, most of the implants in that study were inserted in periodontally compromised sites, which might explain the higher marginal bone loss [19]. Another study by Fügl et al. reported a marginal bone loss of 0.85 mm (standard deviation: 1.37 mm) from insertion to 1-year follow-up in 85 anodically oxidized surface implants [24]. Additionally, Fischer et al. reported a mean marginal bone loss of 1.1 mm (standard deviation: 1.0 mm) at 1 year in 53 anodically oxidized surface implants [25]. Moreover, Glauser et al. reported a marginal bone loss of 1.2 mm (standard deviation: 0.9 mm) after 1 year for anodically oxidized surface implants [26]. A study by Maló et al. using machined surface implants revealed a marginal bone of 1.20 mm (standard deviation: 0.94 mm) at 1-year follow-up [27]. The results obtained in the present study in comparison to the previously reported studies [19,24,25,26,27] suggest that ultra-hydrophilic multi-zone anodized surfaces may play an important role in short-term marginal bone loss, registering significantly lower figures.

The present study reports a 3.2% and 4.8% rate of mechanical complications at implant and patient level, respectively, together with a low percentage of biological complications, with an incidence in only one patient (0.7%) and one implant (0.4%). Malo et al. [19] reported a rate of 0.79% and 0.46% of biological complications at the patient and implant level, respectively, and no mechanical complications were reported. The results between both studies are comparable when considering the biological complications but a higher rate of mechanical complications was reported in the present study. A potential explanation might reside in differences in inclusion and exclusion criteria, given that the previous study [19] excluded bruxers. A previous systematic review of seven studies investigating the influence of bruxism in the outcome of dental implant treatments identified bruxism as a potential risk indicator for mechanical complications [28]. In the present study, the incidence of biological complications was low, but further studies with longer follow-ups are necessary, given the incidence and progression characteristics of peri-implant pathology that require longer periods of exposure.

The limitations of this study include retrospective design, single-center, the short-term follow-up and the lack of a control group with a different implant surface. The study’s strengths include the low loss to follow-up rate that provides a strong internal validity. Future controlled studies with larger samples and longer follow-ups are needed in order to confirm the success obtained at 1-year follow-up. Specifically, the evaluation of the peri-implant complex stability throughout the follow-up is of particular importance, namely soft tissue outcomes such as the bleeding index, probing pocket depths and clinical attachment levels.

## 5. Conclusions

Within the limitations of this study, it is possible to conclude that the use of implants with ultra-hydrophilic multi-zone anodized surfaces for support of single teeth and partial restorations is viable in the short term, with high implant survival rate, low marginal bone loss and low complication rates. Nevertheless, more studies are needed to validate these results.

## Figures and Tables

**Figure 1 jcm-14-00066-f001:**
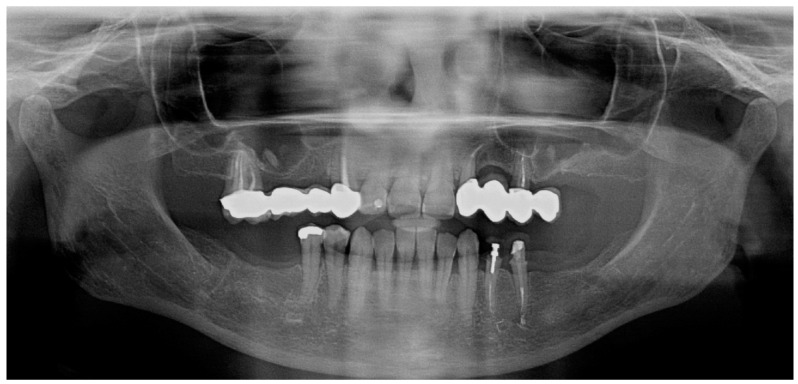
Pre-treatment orthopantomography.

**Figure 2 jcm-14-00066-f002:**
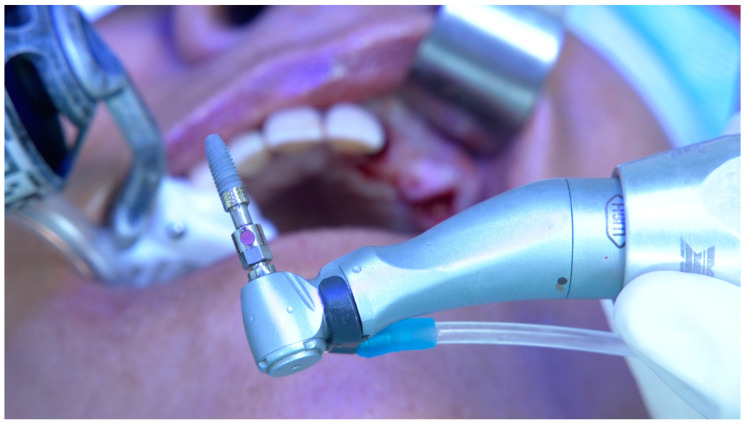
Per-operative extra-oral photograph of the implant to be inserted in position FDI 15.

**Figure 3 jcm-14-00066-f003:**
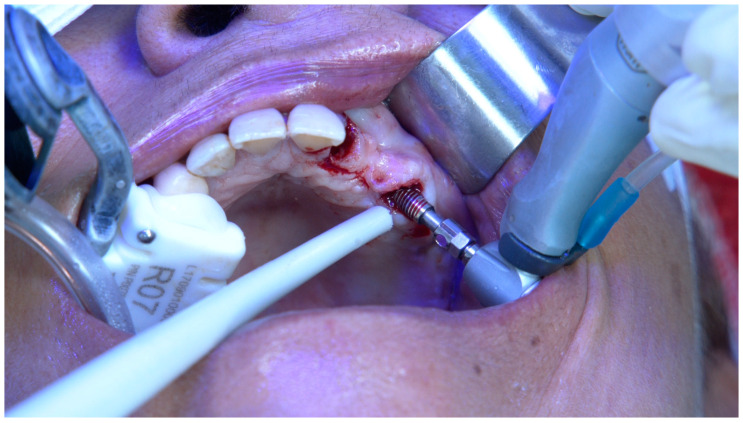
Per-operative intra-oral photograph of the implant insertion in position FDI 15.

**Figure 4 jcm-14-00066-f004:**
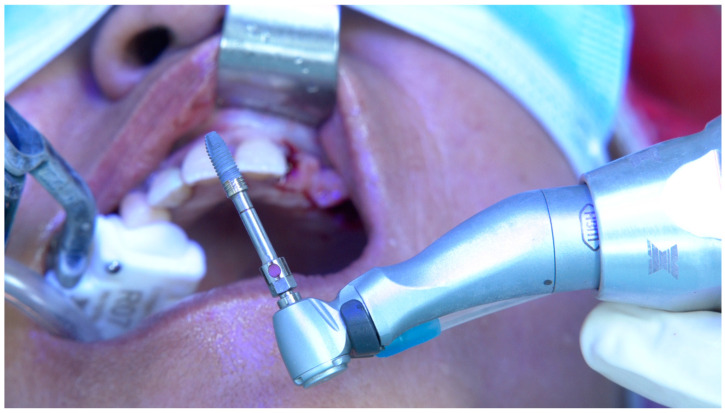
Per-operative extra-oral photograph of the implant to be inserted in position FDI 12.

**Figure 5 jcm-14-00066-f005:**
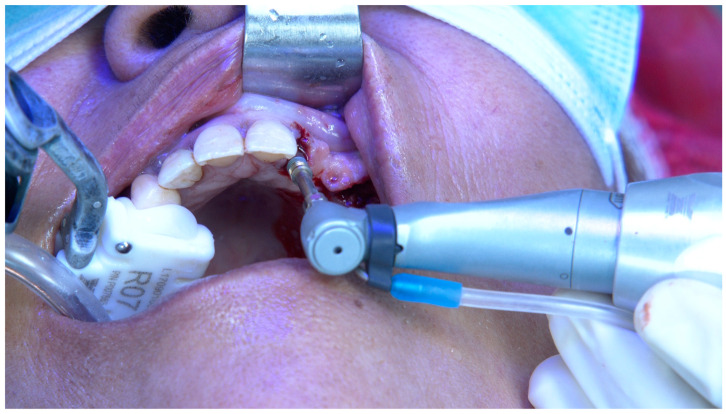
Per-operative intra-oral photograph of the implant insertion in position FDI 12.

**Figure 6 jcm-14-00066-f006:**
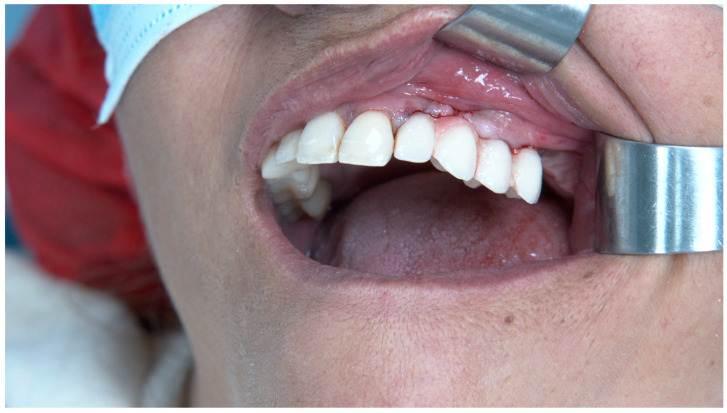
Intra-oral photograph after connection of the immediate provisional prosthesis.

**Figure 7 jcm-14-00066-f007:**
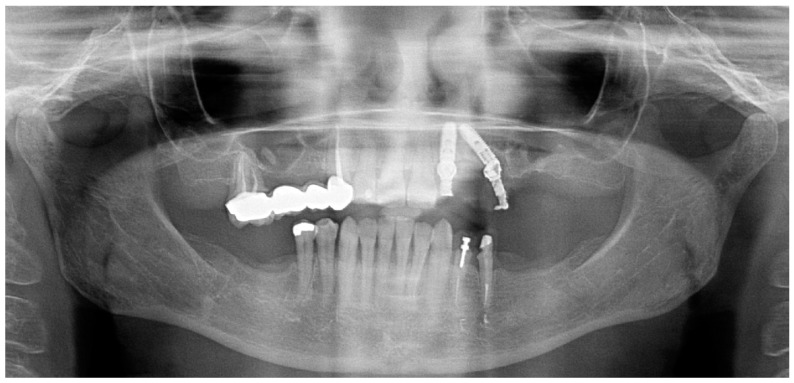
Post-loading orthopantomography.

**Table 1 jcm-14-00066-t001:** Number of implants placed in the maxilla and the mandible, and their position, according to the World Dental Federation.

**Maxilla**															
Position	**17**	**16**	**15**	**14**	**13**	**12**	**11**	**21**	**22**	**23**	**24**	**25**	**26**	**27**	**N**
Number of implants	2	15	11	16	0	3	8	7	5	5	19	17	9	2	119 (47%)
Lost implants	0	0	0	0	0	0	0	0	0	0	0	0	0	0	0 (0%)
**Mandible**															
Position	**47**	**46**	**45**	**44**	**43**	**42**	**41**	**31**	**32**	**33**	**34**	**35**	**36**	**37**	**N**
Number of implants	8	40	16	4	0	0	1	1	0	0	4	7	48	5	134 (52.3%)
Lost implants	0	2	0	0	0	0	0	0	0	0	0	0	0	0	2 (1.5%)

**Table 2 jcm-14-00066-t002:** Distribution of implants by their width and length.

**Width**	**Number of Implants**	**Percentage**
3.75 mm	147	58.1%
4.3 mm	106	41.9%
**Length**	**Number of implants**	**Percentage**
7 mm	9	3.5%
8.5 mm	26	10.3%
10 mm	73	28.9%
11.5 mm	81	32%
13 mm	58	22.9%
15 mm	6	2.4%

**Table 3 jcm-14-00066-t003:** Implant cumulative survival distribution at 1-year follow-up (global sample and according to gender and type of rehabilitation).

**Overall Sample**					
**Time**	Total Number of implants ^a^	Number of failures	Number Lost to follow-up	Survival Rate (%)	Cumulative survival (%)
**0 to 6 months**	253	2	5	99.2	99.2%
**6 months to 1 year**	246	0	10	100%	99.2%
**Distribution by gender ***					
**Male**					
**Time**	Total number of implants ^b^	Number of failures	Number Lost to follow-up	Survival Rate (%)	Cumulative survival (%)
**0 to 6 months**	133	2	5	98.5	98.5
**6 months to 1 year**	126	0	8	100	98.5
**Female**					
**Time**	Total number of implants ^c^	Number of failures	Number Lost to follow-up	Survival Rate (%)	Cumulative survival (%)
**0 to 6 months**	120	0	0	100	100
**6 months to 1 year**	120	0	2	100	100
**Distribution by type of rehabilitation**					
**Single teeth**					
**Time**	Total number of implants ^d^	Number of failures	Number Lost to follow-up	Survival Rate (%)	Cumulative survival (%)
**0 to 6 months**	218	2	5	99.1	99.1
**6 months to 1 year**	211	0	10	100	99.1
**Partial rehabilitations**					
**Time (months)**	Total number of implants ^e^	Number of failures	Number Lost to follow-up	Survival Rate (%)	Cumulative survival (%)
**0 to 6 months**	35	0	0	100	100
**6 months to 1 year**	35	0	0	100	100

Number of patients: a: 145 patients; b: 71 patients; c: 74 patients; d: 134 patients; e: 21 patients. *: *p* = 0.235 (Fisher Exact test).

**Table 4 jcm-14-00066-t004:** Marginal bone loss of implants placed in this study at 1-year follow-up (global sample and according to gender and type of rehabilitation).

**Global Sample**				
Average (standard deviation) in millimeters	0.52 (0.52)			
Number of implants with readable radiographs	201			
Frequencies	N		**%**	
0 mm	70		34.8	
0.1–1 mm	96		47.8	
1.1–2 mm	27		13.4	
2.1–3 mm	6		3	
>3 mm	2		1	
**Distribution according to gender ***				
	Male		Female	
Average (standard deviation) in millimeters	0.60 (0.78)		0.42 (0.55)	
Number of implants with readable radiographs	105		96	
Frequencies	N	%	N	%
0 mm	30	28.6	40	41.7
0.1–1 mm	54	51.4	42	43.8
1.1–2 mm	14	13.3	13	13.5
2.1–3 mm	5	4.8	1	1
>3 mm	2	1.9	0	0
**Distribution according to type of rehabilitation**				
	Single teeth		Partial	
Average (standard deviation) in millimeters	0.52 (0.69)		0.50 (0.62)	
Number of implants with readable radiographs	171		30	
Frequencies	N	%	N	%
0 mm	58	33.9	12	40
0.1–1 mm	85	49.7	11	36.7
1.1–2 mm	21	12.3	6	20
2.1–3 mm	5	2.9	1	3.3
>3 mm	2	1.2	0	0

*: *p* = 0.140 (Mann Whitney U test).

## Data Availability

Data access will be provide by the authors upon reasonable request.

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
