# Peer review of "Single Teeth and Partial Implant Rehabilitations Using Ultra-Hydrophilic Multi-Zone Anodized Surface Implants: A Retrospective Study with 1-Year Follow-Up"

_jcm, 2024, doi:10.3390/jcm14010066_

Round 1
Reviewer 1 Report
Comments and Suggestions for Authors
Dear Authors,
I find your paper entitled "Single teeth and partial implant rehabilitations using ultra-hydrophilic multi-zone anodized surface implants: a retrospective study with 1-year follow-up" interesting, and I consider that it can bring a contribution to the development of the research and practice in the field of dental implants.
Please find below my observations and recommendations for the further improvement of the manuscript.
1. The Introduction section - I consider that the readers need more information concerning the possible complications produced by various types of implant surfaces, the mechanisms of development of such complications, and what has led to the necessity of developing other types of implant surfaces.
2. In the Results section - line 197 -it is mentioned "4 patients", but 3 patients are mentioned in brackets - please clarify.
3. Please consider "gender" instead of "sex", throughout the manuscript.
4. In the Discussion section - please add more comments upon the advantages of ultra-hydrophilic multi-zone anodized surface implants, and how these advantages could be explained.
Author Response
- The Introduction section - I consider that the readers need more information concerning the possible complications produced by various types of implant surfaces, the mechanisms of development of such complications, and what has led to the necessity of developing other types of implant surfaces.
Response: The authors thank the Reviewer’s indication. The possible complications and mechanisms of development of the various implant surfaces were developed.
Changes: Introduction section, lines 48-57.
- In the Results section - line 197 -it is mentioned "4 patients", but 3 patients are mentioned in brackets - please clarify.
Response: Typo. Proof read and corrected.
Changes: Results section, line 195.
- Please consider "gender" instead of "sex", throughout the manuscript.
Response: The authors thank the Reviewer’s indication. Changed as requested.
Changes: Throughout the manuscript.
- In the Discussion section - please add more comments upon the advantages of ultra-hydrophilic multi-zone anodized surface implants, and how these advantages could be explained.
Response: The authors thank the Reviewer’s indication. Comments were added as requested.
Changes: Discussion section, lines 246-251
Reviewer 2 Report
Comments and Suggestions for Authors
This investigation is a paper that presents information for researchers in the field of implant dentistry.
Abstract
This section is correct and reports the main sections of the article
Introduction
The authors reports an update of several aspects of the importance of implant surfaces in the treatment with dental implants. However, it only describes aspects related to a trademark.
The authors reports too times a comercial name of a Company.
The section is short and must be improved with more scientific evidence of this topic.
Reference 8 does not exist as a published article (Pubmed), it is a summary or abstract of a meeting.
The paper aimed to evaluate the short-term outcome of implants of conical connection with anodized ultra-hydrophilic surface with a gradual topography from collar to apex for support of single teeth and partial rehabilitations.
Materials and methods.
The authors are defined each step for the implant procedures.
The authors must updated this section with references of each subsection, specially the surgical and the prosthodontic procedures.
The clinical images of the surgery in Figure 1 need to be improved with higher quality.
Statistical analysis needs to be explained more extensively with the tests used and the significance.
Results.
Results are well structured.
However, the information is only descriptive with numbers, percentages and standard deviation.
The results must be analyzed statistically to assess their statistical significance (chi-square, variance).
Discussion. In this section, the authors evaluates the clinical results according the study. This section shows information for practice applications in implant dentistry
Conclusions.
Authors should incorporate the name of the Conclusions section before the last paragraph of the discussion.
Author Response
Introduction
- The authors reports an update of several aspects of the importance of implant surfaces in the treatment with dental implants. However, it only describes aspects related to a trademark.
Response: The authors thank the Reviewer’s indication. The texto throughout the manuscript was modified to include the surface characteristics and not the trademark name.
Changes: Throughout the manuscript.
- The authors reports too times a comercial name of a Company.
Response: The authors thank the Reviewer’s indication. The commercial name was subtracted from the manuscript to a minimum (on first indication of a material), and as declared on the above point, surface characteristics were used instead.
Changes: Throughout the manuscript.
- The section is short and must be improved with more scientific evidence of this topic.
Response: The authors thank the Reviewer’s suggestion. More scientific evidence was introduced as requested.
Changes: Introduction section, lines 48-57.
- Reference 8 does not exist as a published article (Pubmed), it is a summary or abstract of a meeting.
Response: The authors thank the Reviewer’s indication. However, it is a perfectly sound scientific reference, one of the few on the subject, reviewed by the Board of the European Association for Ossoeintegration to be able to be selected, and published in Clinical Oral Implants Research, a high-impact factor scientific journal.
Changes: None.
Materials and methods.
- The authors must updated this section with references of each subsection, specially the surgical and the prosthodontic procedures.
Response: The authors thank the Reviewer’s indication. The subsections were added as requested.
Changes: Materials and Methods section, lines 79, 88, 97, 118, 156, 168.
- The clinical images of the surgery in Figure 1 need to be improved with higher quality.
Response: The authors thank the Reviewer’s indication. Figure 1 was transformed in figures 1-7 with improved quality.
Changes: Materials and Methods section, lines 132-155
- Statistical analysis needs to be explained more extensively with the tests used and the significance.
Response: The authors thank the Reviewer’s suggestion. Statistical analysis was explained more extensively including the tests used and significance levels.
Changes: Material and Methods section, lines 171-174.
Results.
- The results must be analyzed statistically to assess their statistical significance (chi-square, variance). Response: The authors thank the Reviewer’s indication. The Results were analyzed statistically and their significance assessed for the main outcome measures “implant survival” and “marginal bone loss”.
Line 206, Table 3, line 220 and table 4.
Conclusions.
- Authors should incorporate the name of the Conclusions section before the last paragraph of the discussion.
Response: The authors thank the indication. It was included as requested.
Changes: Line 320.
Round 2
Reviewer 2 Report
Comments and Suggestions for Authors
The authors have revised partially the paper according to the reviewer's recommendations.
Some important recommendations have not been successfully implemented. In fact, the quality of the images ( figures 2-6) has not been improved, which is quite poor for a high-level scientific journal.
Furthermore, the aspects of the methodology related to statistical analysis have not been sufficiently improved.
The use of the different tests is not explained and some tests that do not appear in material and methods are used in the presentation of the results (U-Mann-Whitney).
Author Response
The authors have revised partially the paper according to the reviewer's recommendations.
- Some important recommendations have not been successfully implemented. In fact, the quality of the images ( figures 2-6) has not been improved, which is quite poor for a high-level scientific journal.
Response: The authors thank the Reviewer’s indication and per suggestion of the Editorial Office stating that in pdf format the pictures might be compressed and therefore decrease its quality, the authors have now sent the original pictures to the Editorial office so to comply with the Reviewer’s request. However, the authors pleasantly acknowledge that a high-level scientific journal needs high-level and sound research, which by interpreting the Reviewer’s comments, is something that is not at stake.
Changes: Original files sent to the Editorial Office.
- Furthermore, the aspects of the methodology related to statistical analysis have not been sufficiently improved.
The use of the different tests is not explained and some tests that do not appear in material and methods are used in the presentation of the results (U-Mann-Whitney).
Response: The authors thank the Reviewers comments. The authors acknowledge the mistake on the Materials and Methods section: The t-test was the first test set to use but given that the result of the Kolmogorov-Smirnov test determined that marginal bone loss did not follow a normal distribution, the Mann-Whitney U test (which represents the non-parametric alternative to the parametric t-test) was used. The authors further expanded the comments on the Materials and Methods section on both tests: The variable “implant survival was dichotomized (0: survival; 1: failure) and analyzed using the chi-square test and assuming the applicability conditions (in this case the applicability conditions represent that no cell in the crosstab has an expected frequency of less than 1, and that no more than 20% of the cells have an expected frequency less than 5), the Fisher Exact test. The variable “marginal bone loss” was first evaluated considering its distribution using the Kolmogorov-Smirnov test, and then the difference in distribution evaluated using the non-parametric test alternative to the t-test, the Mann-Whitney U test.
Changes: Materials and Methods section, lines 173-177.